# Learning Adaptive and Temporally Causal Video Tokenization in a 1D Latent Space

## Abstract

We propose AdapTok, an adaptive temporal causal video tokenizer that can flexibly allocate tokens for different frames based on video content. AdapTok is equipped with a block-wise masking strategy that randomly drops tail tokens of each block during training, and a block causal scorer to predict the reconstruction quality of video frames using different numbers of tokens. During inference, an adaptive token allocation strategy based on integer linear programming is further proposed to adjust token usage given predicted scores. Such design allows for sample-wise, content-aware, and temporally dynamic token allocation under a controllable overall budget. Extensive experiments for video reconstruction and generation on UCF-101 and Kinetics-600 demonstrate the effectiveness of our approach. Without additional image data, AdapTok consistently improves reconstruction quality and generation performance under different token budgets, allowing for more scalable and token-efficient generative video modeling.

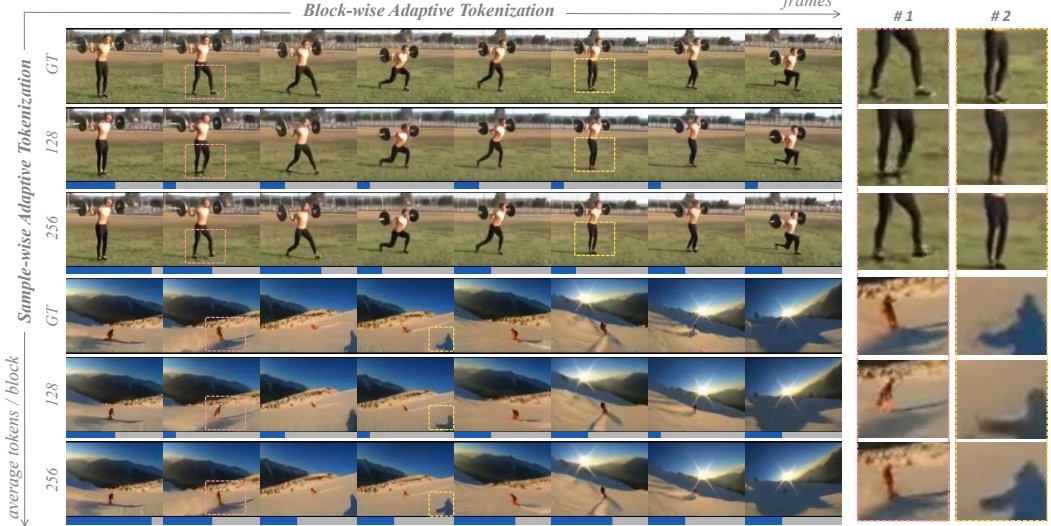

Figure 1: **AdapTok performs adaptive tokenization both temporally and across samples.** Left-to-right shows token allocation adapting over time, top-to-bottom presents sample-wise allocation under different token budgets. Blue bars indicate the tokens counts used per block (i.e., 4 frames).

## 1 Introduction

The vast amount of video data available in the internet and physical world makes efficient video modeling crucial for building visual-centric agents and world models. With the rise of large language models (Radford et al., 2019; Brown et al., 2020; Touvron et al., 2023a), auto-regressive (AR) generative modeling has become a universal paradigm for various modalities, including images and videos (Wang et al., 2024c; Wu et al., 2024; Tian et al., 2024; Li et al., 2024; Jin et al., 2024; Hong et al., 2023).

Consequently, numerous works (Gupta et al., 2022; Wu et al., 2022; Hong et al., 2023; Yan et al., 2021; Ge et al., 2022; Yu et al., 2023; Wang et al., 2024a) have begun exploring video tokenization to quantize spatiotemporally continuous video frames into discrete token sequences, enabling autoregressive generation through causal transformers. Unlike image data, video frames exhibit *temporal causality* despite both being continuous signals. To better model such characteristic, some approaches (Yu et al., 2023; NVIDIA, 2025; Villegas et al., 2022; Wang et al., 2024b) have incorporated temporal causal regularization, supporting online streaming processing of video frames during encoding and decoding, thereby significantly improving throughput efficiency.

However, most existing works (Wang et al., 2024b; NVIDIA, 2025; Wang et al., 2024a) encode different frames using a fixed number of tokens, ignoring the inherent redundancy in video data. While some approaches (Duggal et al., 2024; Koike-Akino & Wang, 2020; Miwa et al., 2025; Bachmann et al., 2025; Shen et al., 2025) have attempted to encode images with variable-length token sequences, few of them is extended to video data. A recent work named ElasticTok (Yan et al., 2024), to the best of our knowledge, made an initial attempt to encode different frames within the same video using varying numbers of tokens. Nevertheless, due to its lack of global planning capabilities and constraints from 2D spatial priors, its token allocation strategy still suffers from imbalances across samples and spatial regions, resulting in suboptimal performance.

From our perspective, an efficient video tokenizer should encompass the following three key characteristics: (1) **temporal causality**, where the encoding and decoding of preceding frames are independent of subsequent frames, thus supporting online streaming processing; (2) **1-D latent token space**, where the token allocation is decoupled from spatial structure, ensuring information density is evenly distributed in space; and (3) **adaptive token allocation**, that adjusts token numbers based on the information of different samples under a given token budget, achieving global optimality.

Motivated by these considerations, we propose AdapTok, a transformer-based framework consisting of a VQ-tokenizer and a causal scorer. During tokenization, the 3D video frame blocks are transformed into 1D latent tokens using a set of learnable latent tokens and a causal block mask is introduced to enable temporal causal modeling. To enable representing videos with variable length latent tokens, we randomly drop tokens at the tail of each block during training. Additionally, a block causal scorer is also trained to model the reconstruction quality of video frames when using token sequences with different token numbers. During inference, we further propose an adaptive token allocation strategy based on integer linear programming (ILP) named IPAL. Given a desired token count, IPAL adjusts token allocation across different samples based on the predicted metrics by trained scorer, thereby optimizing overall reconstruction quality.

Extensive experiments are conducted to validate the effectiveness of our method. Notably, on UCF-101, AdapTok achieves a video reconstruction performance of FVD=28, significantly outperforming existing methods. With our adaptive token allocation strategy, AdapTok achieves Pareto optimality between performance and token count. As shown in Fig.1, AdapTok can adaptively allocate tokens based on video frame content, both temporally and sample-wise, and the reconstruction performance continues to improve as the number of tokens increases. Furthermore, we evaluated video generation performance using AdapTok as a tokenizer on UCF-101 class-conditional generation and Kinetics-600 frame prediction tasks, where our method also demonstrated significant performance gains. Comprehensive ablation studies are also conducted to verify the necessity of each component.

Our contributions can be summarized as follows:

- We propose AdapTok, an adaptive framework designed for video tokenization that simultaneously features temporal causality, a 1D latent token space, and a flexible adaptive token allocation strategy, yielding more compact token representations.

- AdapTok features a novel causal scorer and corresponding adaptive token allocation strategy that dynamically adjusts token allocation across different samples during inference, optimizing overall reconstruction quality.

- AdapTok significantly improves existing video reconstruction performance, achieves Pareto optimality between token allocation quantity and reconstruction performance, and enhances video generation quality of AR models on class-conditional generation and frame prediction tasks.

## 2 RELATED WORK

### 2.1 DISCRETE VIDEO TOKENIZER

Autoregressive (AR) transformers have emerged as powerful generative models across domains, including vision. To handle continuous visual signals, VQ-VAE (Van Den Oord et al., 2017) and its extensions (Esser et al., 2021; Yu et al., 2022; Zheng & Vedaldi, 2023; Zhang et al., 2023; Mentzer et al., 2023) introduce vector quantization to convert images into discrete 2D code sequences. Following works such as TiTok (Yu et al., 2024b) further propose a 1D tokenizer that tokenizes images into compact 1D latent sequences. When applied to videos, early approaches use per-frame tokenization (Gupta et al., 2022; Wu et al., 2022; Hong et al., 2023), where image tokenizers are directly applied without temporal modeling. Block-based tokenization (Yan et al., 2021; Ge et al., 2022; Yu et al., 2023) improves on this by using 3D convolutions to encode short clips with local spatio-temporal context. More recently, LARP (Wang et al., 2024a) introduces holistic tokens to capture global video information. However, similar to most prior approaches, its representations remain non-causal and fixed in length, which limits their applicability to streaming or autoregressive generation tasks. In contrast, we target causal, adaptive video tokenizer that represents videos with variable-length latent tokens, enabling online, low-latency processing while providing flexible representation granularity for downstream tasks.

To address the limitations of non-causal tokenizers, several works have explored causal temporal tokenization architectures. One line of work adopts causal convolutions for autoregressive modeling (Yu et al., 2024a). Building on this, Cosmos-Tokenizer (NVIDIA, 2025) introduces causal convolution layers and causal temporal attention layers for temporal modeling. In parallel, another line of work focuses on transformer-based architectures (Villegas et al., 2022; Wang et al., 2024b), where causality is enforced via attention masks, enabling autoregressive modeling using only past frames. Following such trend, we also adopt a transformer-based causal architecture.

### 2.2 ADAPTIVE TOKENIZER

Recent works have explored adaptive tokenization to allow variable length representations. For image tokenization, ALIT (Duggal et al., 2024) dynamically adjusts the token count by recursively distilling 2D image tokens into 1D latent sequence. Another common strategy is "*tail drop*" (Koike-Akino & Wang, 2020), which applies higher dropout rate on the tail of the latent feature to learn ordered representations. Building on this idea, One-D-Piece (Miwa et al., 2025) and FlexTok (Bachmann et al., 2025) introduce tail-drop regularization for 1D image tokenizers. To support adaptive inference, CAT (Shen et al., 2025) predicts image complexity using captions and large language models (LLMs), assigning each input to one of three compression ratios ($8\times$, $16\times$, $32\times$).

Extending adaptive tokenization to videos, ElasticTok (Yan et al., 2024) applies tail dropping at the end of each block and supports adaptive inference by selecting token counts via a fixed threshold. However, due to the local spatial-temporal nature of these tokens—rather than forming a unified 1D sequence—it is difficult to gather informative content at the head, limiting the effectiveness of tail dropping. Moreover, relying on a fixed threshold prevents token allocation under a specified global budget. In contrast, our AdapTok adopts a 1D transformer that decouples token allocation from spatial structure. We further introduce an online scorer with IPAL for efficient token allocation, to enable globally optimal token allocation under a given token budget.

## 3 METHOD

This work aims to present a flexible video tokenizer that adaptively tokenizes videos both temporally and sample-wise. In pursuit of this goal, we highlight three core designs: an adaptive video tokenizer which allows for representing videos in variable-length (Sec. 3.1), a scoring module (Sec. 3.2), and a token allocation strategy (Sec. 3.3). Complementing this, we also explore its integration in video generation (Sec. 3.4). The overall framework of our method is illustrated in Fig. 2.

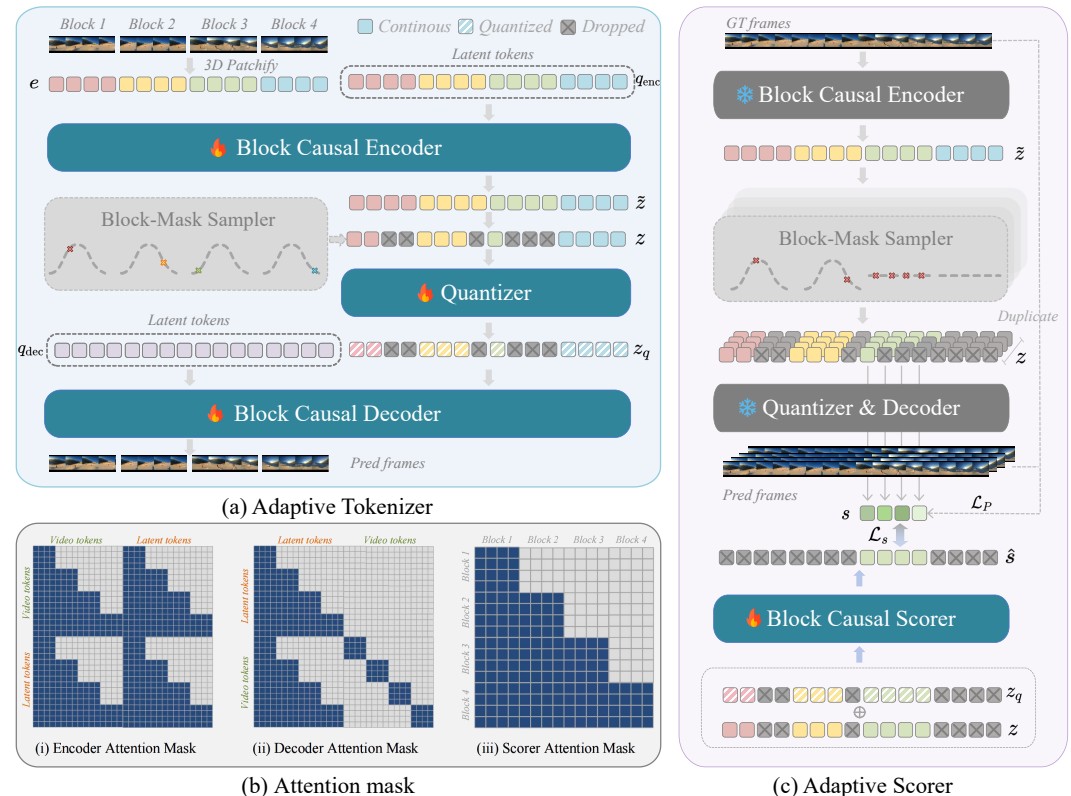

Figure 2: **Overview of the proposed AdapTok framework.** (a) **Adaptive Tokenizer**: composed of a block-causal encoder, a block-wise mask sampler, and a block causal decoder for reconstructing video from adaptively masked latent representations. (b) **Attention Masks**: block-causal attention patterns used in the encoder, decoder, and scorer. (c) **Adaptive Scorer**: top-down illustrates the generation of ground-truth scores - duplicated latent tokens $z$ are masked and decoded into videos, and perceptual loss $\mathcal{L}_P$ is computed as the quality scores $s$. Bottom-up shows the scorer predicting block-wise quality scores $\hat{s}$ from continuous latents $z$ and quantized latents $z_q$.

## 3.1 ADAPTIVE TOKENIZER

### 3.1.1 3D PATCHIFICATION

Given a video input $x \in \mathbb{R}^{T \times H \times W \times 3}$, where $T$ is the number of frames and $H \times W$ is the spatial resolution, we split it into non-overlapping spatio-temporal patches with a patch size of $t \times p \times p$. The video patches are then flattened and projected into patch embeddings $e \in \mathbb{R}^{L \times d}$, where $L = \frac{T}{t} \times \frac{W}{p} \times \frac{H}{p}$ denotes the total number of tokens. The $L$ tokens are then divided into $K$ blocks.

### 3.1.2 BLOCK CAUSAL TRANSFORMER

The block causal transformer employs a causal encoder to extract block-wise representations, a block-mask sampler to randomly sample tokens, an SVQ (Wang et al., 2024a) quantizer for discretization, and a causal decoder to reconstruct videos.

**Block Causal Encoder.** To design an adaptive tokenizer, it is crucial to disentangle the positional dependencies between the latent space and the local spatiotemporal regions. Motivated by Yu et al. (2024b); Wang et al. (2024a), the encoder employs learnable latent tokens to transform patch embeddings into a 1D sequence. Specifically, the latent tokens $q_{enc} \in \mathbb{R}^{N \times d}$ are concatenated with video tokens $e \in \mathbb{R}^{L \times d}$, forming a combined input of $(L + N)$ tokens to a block-causal transformer encoder $\mathcal{E}$. The output corresponding to the latent tokens, denoted as $\tilde{z} = \mathcal{E}(e \oplus q_{enc})_{L:(L+N)}$, serves as the latent representation.

Across blocks, block-causal attention is applied: all tokens in a block (both latent tokens $q_{\text{enc}}$ and video tokens $e$) can only attend to tokens from the same or previous blocks, thereby enforcing a causal dependency structure. The pattern for the encoder attention mask is shown in Fig. 2(b-i).

**Block-mask Sampler.** Inspired by Miwa et al. (2025); Bachmann et al. (2025); Yan et al. (2024), we train AdapTok to learn variable-length codes over a short video block (e.g., 4 frames) by randomly dropping the tail codes for each block during training. Specifically, for each latent block $\tilde{z}_i$, we randomly sample the number of tokens to retain, denoted as $\ell_i$. After sampling $\{\ell_i\}_{i=1}^K$, a binary latent mask $m \in \{0, 1\}^{M \cdot K}$ is constructed, where the first $\ell_i$ values in each of the $K$ blocks are set to 1, and $M$ is the total length of each block that satisfies $N = M \cdot K$.

$$m' = [m_1 \oplus m_2 \oplus \cdots \oplus m_K], \ m_i = [\mathbb{1}_{j \leq \ell_i}]_{j=1}^M. \tag{1}$$

The mask $m'$ is then applied to the latent token sequence $\tilde{z}$, where tokens with $m'_j = 0$ are dropped during training. The resulting sequence, denoted as $z = \text{BlockTailDrop}(\tilde{z} \mid m')$, is then discretized into the quantized feature $z_q$.

**Block Causal Decoder.** The quantized tokens $z_q$ are concatenated with a set of learnable video latent tokens $q_{\text{dec}} \in \mathbb{R}^{L \times d}$, and the combined $(N' + L)$ tokens are fed into a block-causal decoder $\mathcal{D}$, implemented as a transformer. The last $L$ outputs, denoted as $\hat{z} = \mathcal{D}(q_{\text{dec}} \oplus z_q, \mathcal{M}')_{N':(N'+L)}$, are used to reconstruct the video. Here, $N'$ denotes the number of quantized tokens remaining after tail-drop, and $\mathcal{M}'$ is the attention mask applied in the decoder. This mask is derived by applying the dropout mask $m'$ to the original block-causal attention pattern $\mathcal{M}$, as shown in Fig 2(b-ii),

$$\mathcal{M}' = \text{BlockTailDrop}(\mathcal{M} \mid m'). \tag{2}$$

Within this mask, quantized tokens $z_q$ attend only to quantized tokens from the same or preceding blocks, while video latent tokens $q_{\text{dec}}$ attend to all video tokens within the same block and to quantized tokens from the same or earlier blocks.

### 3.1.3 TRAINING

Following previous works Esser et al. (2021); Ge et al. (2022); Wang et al. (2024a), the tokenizer is optimized with a composite objective:

$$\mathcal{L} = \mathcal{L}_R + \mathcal{L}_{VQ} + \mathcal{L}_P + \mathcal{L}_G + \mathcal{L}_{prior}, \tag{3}$$

where $\mathcal{L}_R$ denotes the $L_1$ reconstruction loss, $\mathcal{L}_{VQ}$ the quantizer loss, $\mathcal{L}_P$ the perceptual loss (Zhang et al., 2018), $\mathcal{L}_G$ the adversarial loss (Goodfellow et al., 2014) to enhance visual quality, and $\mathcal{L}_{prior}$ the autoregressive prior loss (Wang et al., 2024a) for modeling the latent sequence.

### 3.2 ADAPTIVE SCORER

To enable adaptive token allocation during inference, we introduce a block-causal scorer that learns to predict reconstruction quality under different token budgets, as illustrated in Fig. 2(c).

**Generating Ground-Truth Scores.** At each iteration, a target block index $q \sim \mathcal{U}(0, K - 1)$ is sampled. We duplicate the latent sequence $z$ and apply a series of latent masks $\{m'_p\}$, each corresponding to a different number of tokens $\ell_q$, ranging from the supported range of encoding lengths. The token lengths $\ell_{i<q}$ for preceding blocks are randomly sampled, while all subsequent blocks are fully masked, i.e., $\ell_{i>q} = 0$. For the $p$-th duplicated sample, the latent mask is given by:

$$m'_p = [m_{p,1} \oplus m_{p,2} \oplus \cdots \oplus m_{p,q} \oplus \mathbf{0}], \ m_{p,i<q} = [\mathbb{1}_{j \leq \ell_i}]_{j=1}^M, \ m_{p,q} = [\mathbb{1}_{j \leq p}]_{j=1}^M. \tag{4}$$

For each $m'_p$, the perceptual loss $\mathcal{L}_P$ over the $q$-th block is computed, used as the quality scores $s$.

**Block Causal Scorer.** The scorer $\mathcal{S}_\phi$, implemented as a transformer encoder with block causal attention, takes both the continuous latent tokens $z$ and the quantized tokens $z_q$ as input. In a single forward pass within a block, it predicts quality scores for all candidate token lengths $\ell_q$, denoted as:

$$\hat{s} = \mathcal{S}_\phi(z \oplus z_q, \mathcal{M}'_s)_{qM:(q+1)M}, \tag{5}$$

where $\mathcal{M}'_s$ is a block-causal attention mask derived from a base pattern $\mathcal{M}_s$ (Fig 2(b-iii)) by applying a dropout mask $m'_s$. The dropout mask $m'_s$ is constructed in a manner similar to $m'_p$ (used for ground-truth generation), except that $\ell_q$ is fixed to the maximum block token length $M$.

The MSE loss $\mathcal{L}_s$ is computed between the predicted score $\hat{s}$ and the ground-truth quality scores $s$.

### 3.3 ADAPTIVE TOKEN ALLOCATION

At inference, we introduce an integer programming-based adaptive allocation strategy (IPAL) that assigns a variable number of tokens to each video block, aiming to optimize the overall reconstruction quality under a global token budget, as detailed in Algorithm 1. Specifically, for each block in a mini-batch $B$, the block-causal scorer $\mathcal{S}_\phi$ predicts a vector of quality scores $\hat{s}_k$ for each sample $k$, across candidate token lengths. A binary variable $b_{kj} \in \{0, 1\}$ indicates whether sample $k$ is assigned $j$ tokens. The ILP is formulated as:

$$\min_b \sum_{k,j} \hat{s}_{kj} \cdot b_{kj} \quad \text{s.t.} \quad \sum_j b_{kj} = 1 \, \forall k, \quad \sum_{k,j} j \cdot b_{kj} = B \cdot N_b, \tag{6}$$

where $B$ is the batch size and $N_b$ is the average number of tokens to be assigned per block.

The objective minimizes the total predicted scores $\hat{s}_{kj}$ (perceptual loss) across a mini-batch, weighted by the binary assignment variables $b_{kj}$. The first constraint ensures that each sample selects exactly one token length, while the second constraint enforces the total token budget across the batch. The optimal assignment $b^*$ yields the final token count for each sample: $n_k = \sum_j j \cdot b_{kj}^*$.

---

**Algorithm 1:** Integer Linear Programming

1 **Inputs:** Video $x$, Block idx $q$;
2 **Hyperparameters:** Average Token $N_b$,
   tokens per block $M$, batch size $B$;
3 $z = \mathcal{E}(x)$, $z_q = \mathcal{Q}(z)$;
4 $\hat{s} = \mathcal{S}_\phi(z, z_q)_{qM:(q+1)M}$;
   // solve the ILP
5 **Define:** $b_{kj} \in \{0, 1\}$ for binary score bins;
6 **Objective:** $\mathcal{L} = \min_b \sum_{k,j} \hat{s}_{kj} b_{kj}$;
7    s.t. $\sum_j b_{kj} = 1$, $\forall k$;
8    $\sum_{k,j} j \cdot b_{kj} = B \cdot N_b$;
9 $b^* = \text{ILPSolve}(\mathcal{L})$;
10 $n_k = \sum_j j \cdot b_{kj}^*$;
11 **Return:** assigned tokens $n_k$;

---

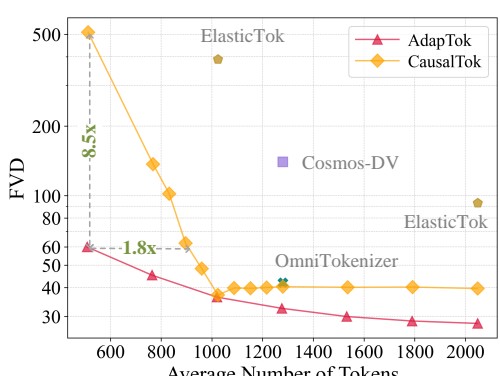

Figure 3: **Comparison of rFVD by Token Length.** AdapTok achieves better rFVD with fewer tokens than existing causal tokenizers.

### 3.4 VIDEO GENERATION

Given a sequence of adaptively allocated tokens from the scorer with IPAL, an end-of-block token `<EOB>` is appended to the end of each block. The resulting concatenated sequence is then fed into a Llama-style transformer (Touvron et al., 2023a;b) for autoregressive generation. Formally, let $y = (y_1, y_2, \cdots, y_S)$ denote the multi-block adaptive sequence. The model is trained to model the probability distribution by maximize the likelihood of each token $y_i$ autoregressively using a standard cross-entropy loss:

$$\mathcal{L} = -\sum_{i=1}^{S} \log P(\hat{y}_i | c, y_{1:i-1}; \theta), \tag{7}$$

where $c$ denotes the conditions, e.g., class labels for class-conditional generation or context tokens for frame prediction, and $\theta$ represents the trainable parameters of the transformer.

## 4 EXPERIMENTS

### 4.1 EXPERIMENTAL SETUPS

**Datasets.** We train the model and evaluate its performance on two standard datasets: video reconstruction and class-conditional generation on UCF-101 (Soomro et al., 2012) and frame prediction on Kinetics-600 (Carreira et al., 2018) with 5-frame conditioning.

**Implementation details.** Following Ge et al. (2022); Yu et al. (2024a), we adopt $16 \times 128 \times 128$ video clips for both training and evaluation. Patch embeddings are extracted with a patch size of

Table 1: **Comparison of Video Reconstruction FVD on UCF-101.** All models use a causal visual tokenizer. **Bold** and underline indicate the best and the second best performance, respectively. † denotes models trained using the same dataset and receipe as ours.

| Method | Data size | Codebook | Tokens | rFVD ↓ |
|---|---|---|---|---|
| OmniTokenizer (Wang et al., 2024b) | 1.4M | 8,192 | 1,280 | 42 |
| ElasticTok (Yan et al., 2024) | 356M | 64,000 | 1,024 | 390 |
| ElasticTok (Yan et al., 2024) | 356M | 64,000 | 2,048 | 93 |
| Cosmos-Tokenizer-DV (NVIDIA, 2025) | 100M | 64,000 | 1,280 | 140 |
| OmniTokenizer † (Wang et al., 2024b) | <0.5M | 8,192 | 1,280 | 94 |
| ElasticTok † (Yan et al., 2024) | <0.5M | 64,000 | 1,022 | 230 |
| CausalTok † | <0.5M | 8,192 | 1,024 | 37 |
| AdapTok (Ours) | <0.5M | 8,192 | 512 | 60 |
| AdapTok (Ours) | <0.5M | 8,192 | 1,024 | 36 |
| AdapTok (Ours) | <0.5M | 8,192 | 2,048 | **28** |

Table 2: **Class-conditional generation results on UCF-101 and frame prediciton results on Kinetics-600.** ‡ denotes using pretrained image tokenizer.

| Method | Data size | Params | gFVD ↓ | |
|---|---|---|---|---|
| | | | K600 | UCF |
| Phenaki (Villegas et al., 2022) | 465M | 1.8B | 36.4 | / |
| MAGVIT-AR (Yu et al., 2023) | 12M | 306M | / | 265 |
| MAGVIT-v2-AR (Yu et al., 2024a) | 1.8M | 840M | / | 109 |
| OmniTokenizer (Wang et al., 2024b) | 1.4M | 650M | 32.9 | 191 |
| CogVideo ‡ (Hong et al., 2023) | 5.4M | 9.4B | 109.2 | 626 |
| Video-LaVIT ‡ (Jin et al., 2024) | 10M | 7B | / | 281 |
| TATS (Ge et al., 2022) | <0.5M | 321M | / | 332 |
| CausalTok | <0.5M | 633M | / | 80 |
| AdapTok-AR (Ours) | <0.5M | 633M | **11** | **67** |

$4 \times 8 \times 8$, resulting in $L = 1024$ tokens per video. A sequence of latent tokens $q_{enc}$ is used to extract these patch embeddings, and is partitioned into $K = 4$ blocks, each containing $M = 512$ tokens.

For adaptive token sampling, we employ a block-wise mask sampler. In each block, the token number is sampled from a truncated Gaussian distribution with mean $\mu = 256$, standard deviation $\sigma = 128$, and bounded between $M_{min} = 32$ and $M_{max} = 512$.

AdapTok is trained on UCF-101 and Kinetics-600 for 250 epochs with a batch-size of 128. Adam (Kingma & Ba, 2014) is adopt as the optimizer with hyperparameters $\beta_1 = 0.5$ and $\beta_2 = 0.9$. The learning rate is linearly warmed up to $10^{-4}$ and then decayed to $10^{-6}$ using a cosine scheduler.

Fréchet Video Distance (FVD) (Unterthiner et al., 2018) is adopt as the primary metric for reconstruction and generation. Additionally, we report PSNR and LPIPS for video reconstruction.

## 4.2 MAIN RESULTS

**Video Reconstruction.** We first evaluate the video reconstruction performance of AdapTok on UCF-101, with results summarized in Tab. 1. In particular, we implement a strong baseline named CausalTok that shares the same block-causal architecture as AdapTok, but applies a standard non-adaptive pipeline. To ensure fair comparison, we also reproduced previous works (i.e., ElasticTok and OmniTokenizer) using the same dataset and receipe as ours. With fewer training data and latent tokens, AdapTok significantly outperforms existing causal video tokenizers, achieving an rFVD of 28 with 2048 tokens and 36 with 1024 tokens.

To further evaluate efficiency under varying token budgets, we compare AdapTok with several causal tokenizer baselines. As shown in Fig. 3, AdapTok consistently achieves lower rFVD across different token counts. Remarkably, it achieves an FVD of 60 using only 512 tokens, outperforming most baselines, and achieves comparable performance to CausalTok while using $1.8\times$ fewer tokens.

Table 4: **Ablation on adaptive training and inference.**

| Tokens | Sampler | Scorer | rFVD ↓ | PSNR ↑ | LPIPS ↓ |
|---|---|---|---|---|---|
| | ✗ | ✗ | 37.13 | **25.92** | **0.111** |
| 1024 | ✓ | ✗ | 38.79 | 25.29 | 0.122 |
| | ✓ | ✓ | **36.36** | 25.72 | 0.114 |
| | ✗ | ✗ | 509.95 | 14.38 | 0.368 |
| 512 | ✓ | ✗ | 121.88 | 22.89 | 0.170 |
| | ✓ | ✓ | **59.96** | **24.06** | **0.144** |

Table 5: **Comparison of inference latency.** AdapTok achieves **11×** lower latency than ElasticTok.

| Method | Time (ms/video) |
|---|---|
| ElasticTok | 571.7 |
| AdapTok | 50.9 |

Table 6: **Ablation on token allocation strategies.**

| Method | rFVD ↓ | PSNR ↑ | LPIPS ↓ |
|---|---|---|---|
| Fixed | 38.79 | 25.29 | 0.122 |
| BiThr | 42.12 | 25.23 | 0.120 |
| BiDelta | 38.13 | 25.65 | 0.115 |
| ILP | **36.36** | **25.72** | **0.114** |

Table 7: **Comparison on scoring metrics.**

| Metric | rFVD ↓ | PSNR ↑ | LPIPS ↓ |
|---|---|---|---|
| SSIM | 37.12 | 25.74 | 0.115 |
| PSNR | 36.56 | **25.84** | 0.115 |
| MSE | 36.97 | 25.75 | 0.115 |
| Perceptual | **36.28** | 25.72 | **0.113** |

**Video Generation.** For video generation, we use AdapTok to tokenize videos and train a Llama-style model to autoregressively generate token sequences. As reported in Tab. 2, our model achieves competitive performance on both class-conditional generation and frame prediction. With only 633M parameters, AdapTok-AR achieves a gFVD of 11 on Kinetics-600 and 67 on UCF-101. Notably, with similar reconstruction performance, AdapTok achieves better generation results than CausalTok, demonstrating the advantage of adaptively allocating tokens for different videos.

**Model Scaling.** To further investigate the impact of model size on performance, we trained models across 3 different sizes: AdapTok-S, AdapTok-L, and AdapTok-XL. As shown in Tab. 3, the reconstruction quality improves consistently with increasing model size. Such results indicate that AdapTok benefits from the increased model capacity, highlighting the scaling potential of our method.

Table 3: **Performance of AdapTok with different model sizes.** Continuous improvement is achieved as the model size increases.

| Model | Params | rFVD ↓ | PSNR ↑ | LPIPS ↓ |
|---|---|---|---|---|
| AdapTok-S | 59M | 87.32 | 24.23 | 0.151 |
| AdapTok-L | 259M | 36.36 | 25.72 | 0.144 |
| AdapTok-XL | 913M | 32.43 | 26.29 | 0.103 |

## 4.3 ABLATION STUDIES

**Adaptive training and inference paradigms.** AdapTok leverages two adaptive mechanisms: a block-mask sampler for training and an adaptive scorer for inference. The block-mask sampler enables the model to learn from variable token lengths by randomly sampling token counts per block, while the adaptive scorer allocates tokens based on reconstruction quality scores during inference. As shown in Tab. 4, leveraging both mechanisms results in the best performance, achieving the lowest rFVD of 36.36. The performance gain becomes more pronounced with lower token budgets in inference, highlighting the importance of our adaptive mechanisms for efficient video representation.

**Token allocation strategies.** We explore several token allocation strategies: **1)** *Fixed token allocation*, which assigns a fixed token count to all blocks; **2)** *Score-threshold binary search (BiThr)*, which binary searches for a global score threshold and selects the minimal token count per block with predicted score below it; **3)** *Delta-score binary search (BiDelta)*, which selects the minimal token count where the score improvement between consecutive lengths drops below a binary searched delta; and **4)** *Integer Linear Programming (ILP)*, which jointly optimizes token allocation across a mini-batch under an average token budget. As reported in Tab. 6, the ILP strategy outperforms the other methods across all metrics. Fig. 4 further compares the different token allocation strategies across varying token lengths, demonstrating that the ILP-based allocation strategy consistently outperforms the alternatives.

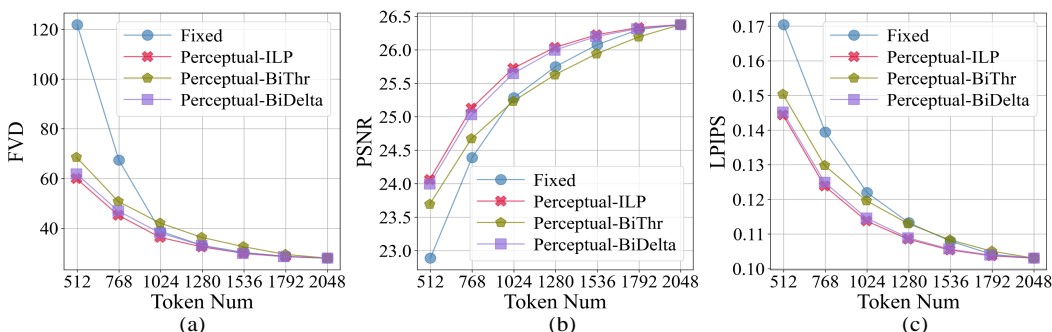

Figure 4: **Comparison of token allocation strategies.** ILP achieves the best performance across all metrics: (a) FVD, (b) PSNR, and (c) LPIPS.

**Token allocation scoring metrics.** We also investigate various scoring metrics for the token allocation, including per-sample SSIM, PSNR, MSE, and perceptual loss. As shown in Tab. 7, each scoring metric performs best on its corresponding evaluation metric, e.g., PSNR score yields the highest PSNR. Notably, using perceptual loss as the scoring metric also leads to superior performance on other evaluation metrics, such as rFVD, indicating its stronger correlation with overall perceptual quality. For a comparison under varying token lengths, please refer to the Appendix C.

**Computational costs analysis.** The comparison of computational cost between ElasticTok and ours is reported in Tab. 5. Thanks to the design of the scorer and IPAL, our method achieves significantly lower inference latency (**11× faster**) compared to ElasticTok. The time proportion of IPAL also remains stable, regardless of varying batch sizes and average token numbers. Please refer to the Appendix C for more detailed experiment results and analyses.

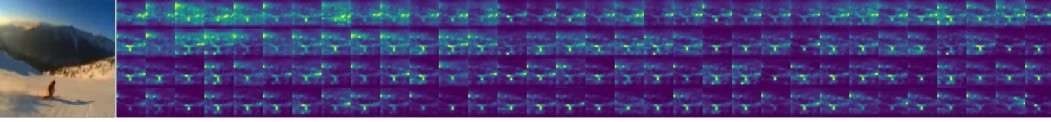

Figure 5: **Attention maps for latent tokens.** Each map shows the attention distribution of a token over spatial patches, with tokens ordered from top-left to bottom-right.

### 4.4 VISUALIZATIONS

**Head tokens encode global information.** The tail token drop strategy encourages the head tokens to capture more global information. As shown in Fig.5, early tokens capture global context, while later tokens focus on local details. Please refer to the Appendix D for more visualization results.

## 5 CONCLUSION

In this paper, we propose AdapTok, an adaptive video tokenizer designed with temporal causality within a unified 1D latent token space. AdapTok incorporates a novel causal scorer and an ILP-based allocation strategy, IPAL, which dynamically adjusts token usage both temporally and sample-wise during inference. Extensive experiments demonstrate that AdapTok not only achieves superior reconstruction quality compared to existing causal video tokenizers, but also provides a favorable trade-off between performance and token budgets, and improves video generation in both class-conditional and frame prediction tasks.

**Limitations and Future work.** In this work, we focus on the design of a sample-wise, content-aware, and temporally adaptive tokenization framework. Our current implementation adopts a discrete tokenizer based on VQ-VAEs (Van Den Oord et al., 2017; Esser et al., 2021), and future work may explore continuous alternatives to validate the generality of the proposed framework. In addition, our model is trained on less than 0.5M publicly available videos due to limited computational resources. Future efforts will focus on scaling up model capacity and dataset diversity to improve generalization and applicability across broader domains.

ETHICS STATEMENT

This work uses only publicly available datasets that have been rigorously filtered to mitigate potential biases and ethical concerns.

REPRODUCIBILITY STATEMENT

The implementation details of our method are fully described in Sec. 4.1 and Appendix. A, including both hyper-parameter settings and training costs. Additionally, the pseudo-code for IPAL and details of other token allocation strategies are provided in Algorithm 1 and Appendix. B. The source code and checkpoints will be released to enable reproduction of the main results presented in this paper.

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

## TECHNICAL APPENDICES AND SUPPLEMENTARY MATERIAL

In the supplementary materials, we provide the following additional details:

- **Sec. A** The comprehensive **hyper-parameters and training costs** for AdapTok.
- **Sec. B** The detailed implementation for the quantization method and several other **adaptive inference strategies**, such as Fixed, BiThr and BiDelta token allocation strategies.
- **Sec. C** More **ablation experiments** on the token scoring metrics, mini-batch size, block number, and average token count.
- **Sec. D** More **qualitative visualizations**, including
    - **Sec. D.1** More adaptive reconstruction results to demonstrate that the necessity of tokenizing in 1D latent space (Fig. 7), as well as AdapTok's ability to perform content-aware (Fig. 8) and temporally dynamic (Fig. 9) token allocation;
    - **Sec. D.2** Video generation results on UCF-101 class-conditional generation (Fig. 10) and Kinetics-600 frame prediction (Fig. 11);
    - **Sec. D.3** More attention maps for latent tokens (Fig. 12).
    - **Sec. D.4** The visualization of scorer predictions and corresponding reconstruction results under varying token counts (Fig. 13).
- **Sec. E** The Use of Large Language Models (LLMs).

## A   IMPLEMENTATION DETAILS

The detailed training hyper-parameter settings for the AdapTok, Scorer, AdapTok-AR (class-conditional generation on UCF-101) and AdapTok-FP (frame prediciton on Kinetics-600) are reported in Table 8. The architectural configurations of the different AdapTok variants are listed in Table 9.

Table 8: Hyper-parameters for AdapTok models.

|  | AdapTok | Scorer | AdapTok-AR | AdapTok-FP |
|---|---|---|---|---|
| *Model parameters* | | | | |
| Parameters | 195M | 89M | 633M | 633M |
| Frame Resolution | $16 \times 128 \times 128$ | $16 \times 128 \times 128$ | $16 \times 128 \times 128$ | $16 \times 128 \times 128$ |
| Patch Size | $4 \times 8 \times 8$ | $4 \times 8 \times 8$ | $4 \times 8 \times 8$ | $4 \times 8 \times 8$ |
| Hidden Size | 768 | 768 | 1280 | 1280 |
| Transformer Layers | 12 | 12 | 30 | 30 |
| *Training* | | | | |
| Optimizer | Adam | Adam | AdamW | AdamW |
| Learning rate | $1e^{-4}$ | $1e^{-4}$ | $6e^{-4}$ | $6e^{-4}$ |
| Beta1 | 0.5 | 0.5 | 0.9 | 0.9 |
| Beta2 | 0.9 | 0.9 | 0.95 | 0.95 |
| Weight decay | 0 | 0 | 0.05 | 0.05 |
| Scheduler type | cosine | cosine | cosine | cosine |
| Warmup epochs | 8 | 8 | 4 | 1 |
| Batch size | 128 | 128 | 64 | 64 |
| Epochs | 250 | 20 | 3000 | 75 |
| GPUs | 32 | 32 | 8 | 16 |
| Training Time | 112h | 9h | 59h | 55h |

Table 9: Model configurations of AdapTok variants.

| Model | Hidden Size | Depth | Heads | Parameters |
|---|---|---|---|---|
| AdapTok-S | 512 | 6 | 8 | 59M |
| AdapTok-L | 768 | 12 | 12 | 259M |
| AdapTok-XL | 1024 | 24 | 16 | 913M |

## B   MORE TECHNICAL DETAILS

### B.1   QUANTIZATION METHOD

Following Wang et al. (2024a), a stochastic vector quantization (SVQ) is adopted to the quantizer $\mathcal{Q}$. Similar to VQ, a codebook $C \in \mathbb{R}^{c \times d'}$ containing $c$ codes is maintained. Given a visual feature $z$, the cosine similarity with all code vectors in $C$ is computed, followed by a softmax to obtain a probability distribution. An index $x_{\text{Ind}}$ is then sampled from this distribution via a categorical distribution:

$$x_{\text{Ind}} \sim \text{Categorical} \left( \text{softmax} \left( \left\{ \frac{v \cdot C_i}{\|v\| \|C_i\|} \right\}_{i=1}^{c} \right) \right). \tag{8}$$

Given the sampled index, the quantized feature $z_q$ is retrieved from the codebook, i.e., $z_q = C_{x_{\text{Ind}}}$. To enable differentiable training, the straight-through estimator Bengio et al. (2013) is applied.

### B.2   ADAPTIVE INFERENCE

In addition to the Integer Linear Programming (ILP) method detailed in the main paper, we provide further details for the other three token allocation methods, including Fixed, BiThr and BiDelta.

**Fixed token allocation** uses the same number of tokens across all video samples and blocks. The latent mask is given by:

$$m' = [m_1 \oplus m_2 \oplus \cdots \oplus m_K], \ m_i = [\mathbb{1}_{j \leq N_b}]_{j=1}^{M}. \tag{9}$$

where $N_b$ is the fixed number of tokens allocated per block.

**Score-threshold binary search (BiThr)** binary searches a global score threshold which assigns the minimum token counts that satisfy a desired video quality. Specifically, given a score threshold $s_i$, the token counts are assigned by selecting the first position where the score drops below $s_i$. The threshold is iteratively updated via binary search until the average token count matches the target value, as detailed in Algorithm 2.

**Delta-score binary search (BiDelta)** follows the same binary search procedure as BiThr, but operates on delta scores instead of raw scores. These delta scores, computed using a difference function (see Algorithm 2), quantify the marginal gain in perceptual quality from adding each token. The search aims to find a threshold over these deltas such that the resulting token allocation meets the desired token count.

---

**Algorithm 2:** Binary Search

1 **Inputs:** Video $x$, Block idx $i$;
2 **Hyperparameters:** Average Token $N_b$, tokens per block $M$, max iterations $K$;
3 $z = \mathcal{E}(x), z_q = \mathcal{Q}(z)$;
4 $s = \begin{cases} \mathcal{S}_\phi(z, z_q)_{iM:(i+1)M}, & \text{(BiThr)} \\ \mathcal{S}_\phi(z, z_q)_{iM:(iM+M-1)} - \mathcal{S}_\phi(z, z_q)_{(iM+1):(iM+M)}, & \text{(BiDelta)} \end{cases}$;
5 $s_{max}, s_{min} = \max(s), \min(s)$;
6 **for** $k = 1, \cdots, K$ **do**
7 $\quad s_{mid} = (s_{max} + s_{min})/2$;
8 $\quad n_k = \text{argmax}(s < s_{min})$;
9 $\quad$ **if** $\text{mean}(n_k) > N_b$ **then**
10 $\quad\quad s_{min} = s_{mid}$;
11 $\quad$ **else**
12 $\quad\quad s_{max} = s_{mid}$;
13 **Return:** assigned tokens $n_k$;

---

## C ADDITIONAL EXPERIMENTS

**Ablation on token allocation scoring metrics.** Fig. 6 presents detailed comparisons of token allocation under varying token lengths for different scoring metrics. Each metric achieves the best performance on its corresponding evaluation metric, while perceptual loss consistently yields strong results across multiple metrics, demonstrating its effectiveness for guiding adaptive token allocation.

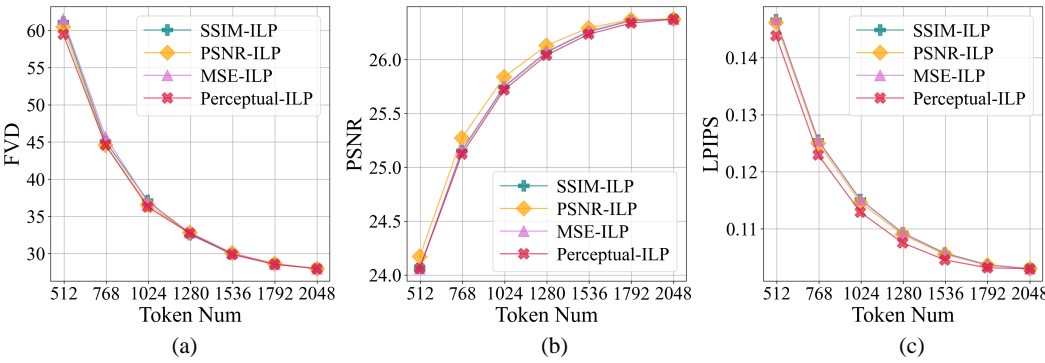

Figure 6: **Comparison of scoring metrics.** Using perceptual loss as the scoring metric achieves better overall performance, especially on FVD and LPIPS.

**Further analysis on computational costs.** Although IPAL is based on ILP, its actual runtime overhead is not significant. Table 10-12 shows the runtime scalability of IPAL *w.r.t.* mini-batch size, the number of blocks and average token numbers. Specifically, across batch sizes ranging from 8 to 1024, FVD remains stable between 36 and 37, while IPAL accounts for only about 15% of total inference time. The allocation time also remains stable with varying average token numbers, while it increases moderately as the number of blocks grows. These results suggest that our method is efficient, scalable, and practical for real-world deployment scenarios.

Table 10: **Ablation on mini-batch size with an average token of 512.**

| Batch size | FVD ↓ | PSNR ↑ | LPIPS ↓ | Time (ms/video) | | IPAL Time Proportion (%) |
|------------|-------|--------|---------|-------|------|-------------------------|
| | | | | Total | IPAL | |
| 8 | 60.39 | 24.01 | 0.146 | 66.3 | 28.0 | 42.2 |
| 16 | 60.14 | 24.04 | 0.145 | 53.7 | 16.7 | 31.2 |
| 64 | 59.96 | 24.06 | 0.144 | 45.2 | 8.6 | 19.1 |
| 128 | 60.61 | 24.07 | 0.144 | 44.1 | 7.0 | 15.8 |
| 256 | 61.48 | 24.08 | 0.144 | 44.0 | 6.7 | 15.2 |
| 512 | 61.44 | 24.08 | 0.144 | 43.0 | 6.3 | 14.8 |
| 1024 | 61.37 | 24.09 | 0.144 | 44.1 | 6.8 | 15.5 |

Table 11: **Ablation on mini-batch size with an average token of 1024.**

| Batch size | FVD ↓ | PSNR ↑ | LPIPS ↓ | Time (ms/video) | | IPAL Time Proportion (%) |
|------------|-------|--------|---------|-------|------|-------------------------|
| | | | | Total | IPAL | |
| 8 | 36.55 | 25.68 | 0.115 | 69.0 | 28.1 | 40.7 |
| 16 | 36.80 | 25.71 | 0.114 | 56.8 | 16.9 | 29.8 |
| 64 | 36.36 | 25.72 | 0.114 | 48.1 | 8.5 | 17.8 |
| 128 | 36.52 | 25.73 | 0.114 | 46.7 | 7.2 | 15.4 |
| 256 | 37.02 | 25.74 | 0.114 | 47.4 | 6.9 | 14.6 |
| 512 | 36.93 | 25.74 | 0.113 | 46.9 | 7.0 | 14.8 |
| 1024 | 37.01 | 25.75 | 0.113 | 47.3 | 7.5 | 15.8 |

# D    MORE VISUALIZATIONS

## D.1    VIDEO RECONSTRUCTION

**1D latent token space matters.** In Fig. 7, we visualize reconstruction results under different token budgets. Compared to ElasticTok, which relies on local 2D spatial tokens and tends to produce block-like artifacts, AdapTok leverages 1D latent space that enables a coarse-to-fine reconstruction process: early tokens capture global structure, while later tokens refine local details.

**Content-aware allocation.** As shown in Fig. 8, AdapTok adaptively allocates tokens based on the visual complexity of each scene. Static or low-motion segments receive fewer tokens, while dynamic or visually complex regions are assigned more, enabling efficient token usage without compromising important content.

**Temporal dynamics.** Fig. 9 illustrates how AdapTok adaptively allocates tokens over time. When scene changes, more tokens are assigned to capture the transition, while fewer tokens are used in stable or redundant segments.

## D.2    VIDEO GENERATION

We present class conditional generation results on UCF-101 in Fig. 10 and frame prediction results on Kinetics-600 in Fig. 11.

Table 12: **Runtime ablation of IPAL on block and token counts.** (a) Ablation on block numbers. (b) Ablation on average token counts.

| (a) Number of blocks. | | | (b) Average tokens counts. | | |
|---|---|---|---|---|---|
| Blocks | Tokens | IPAL Time (ms/video) | Blocks | Tokens | IPAL Time (ms/video) |
| 1 | 1024 | 7.9 | 4 | 32 | 7.8 |
| 2 | 1024 | 7.1 | 4 | 128 | 8.5 |
| 4 | 1024 | 8.0 | 4 | 512 | 8.3 |
| 8 | 1024 | 10.6 | 4 | 1024 | 8.0 |

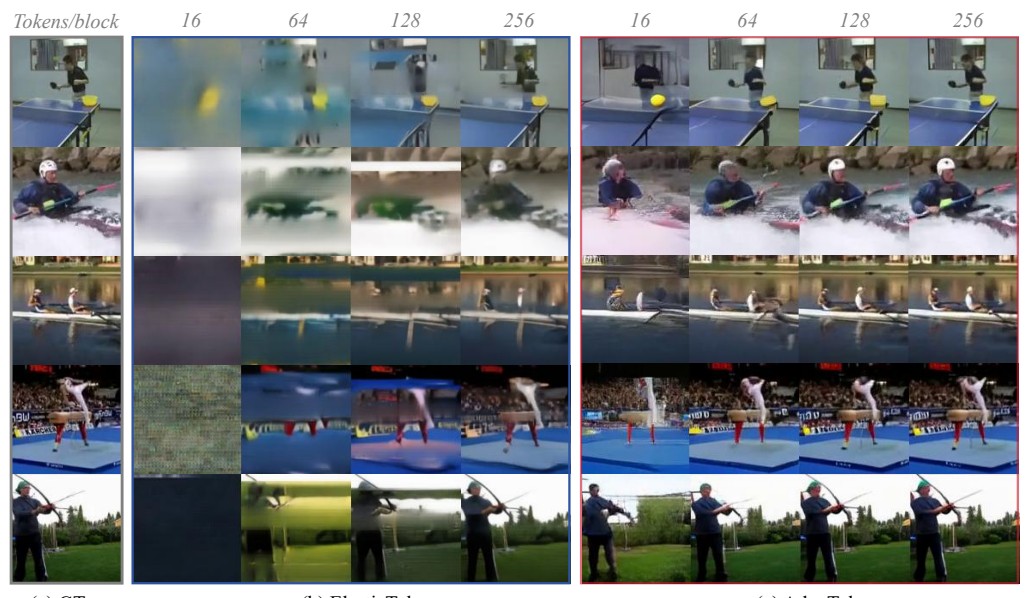

(a) GT        (b) ElasticTok        (c) AdapTok

Figure 7: **Reconstruction comparison with ElasticTok Yan et al. (2024) under varying token budgets.** AdapTok reconstructs videos in a coarse-to-fine manner, where early tokens capture global structure and later ones refine local details, while ElasticTok suffers from blocky artifacts.

## D.3 TOKEN CONTRIBUTION ANALYSIS

Fig. 12 provides additional attention maps of latent tokens from various samples. These further demonstrate that early tokens tend to attend broadly to capture global context, while mid-to-late tokens focus more on local details.

## D.4 SCORER PREDICTIONS

Fig. 13 illustrates the scorer's prediction quality and the effect of increasing token counts on reconstruction. As shown in Fig. 13a, the predicted scores closely match the ground truth scores (perceptual loss) across all blocks. As more tokens are allocated within each block (moving along the x-axis), the score decreases, indicating improved reconstruction quality. This trend is visually confirmed in Fig. 13b, where reconstructions become progressively more accurate as token counts increase.

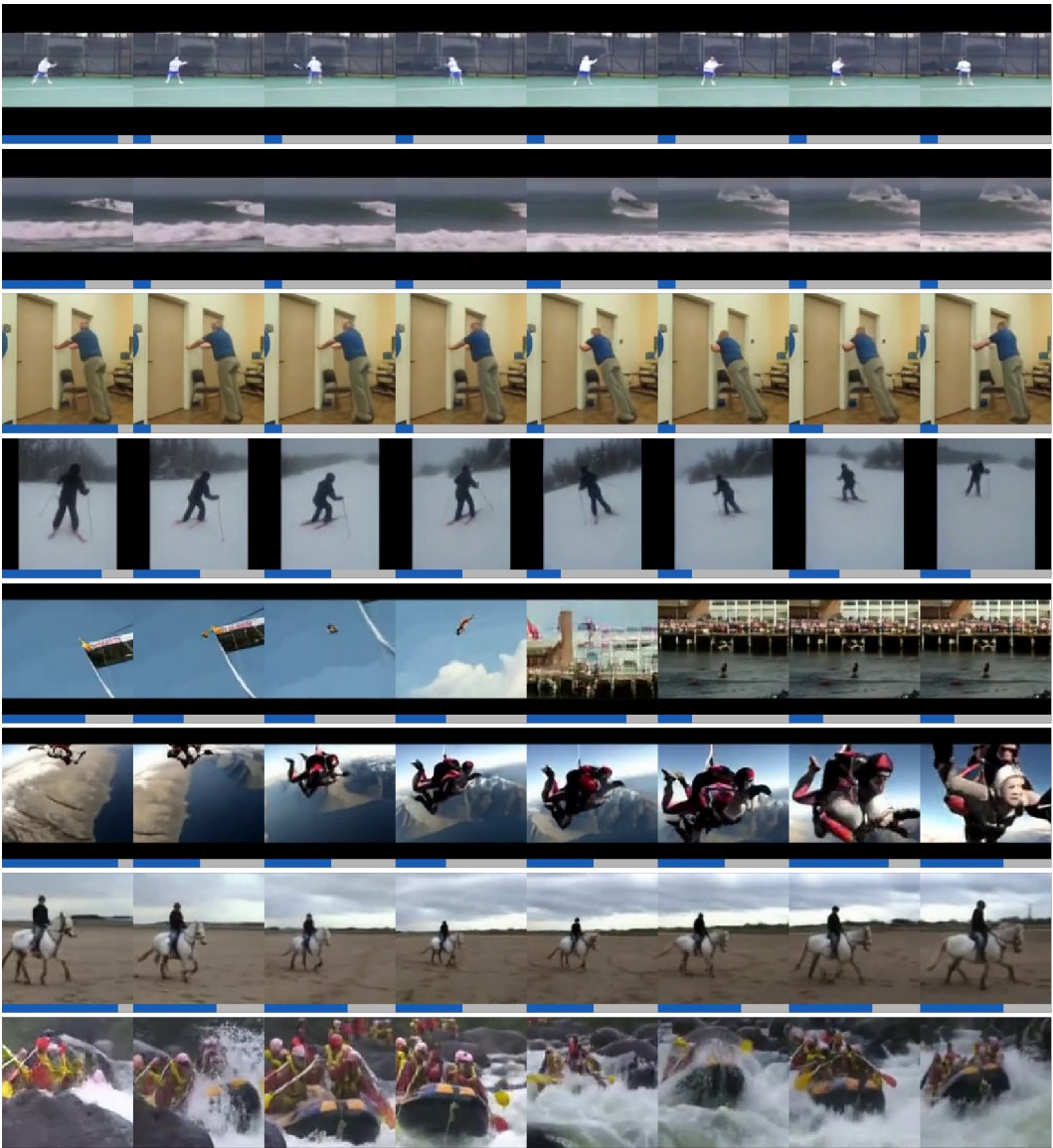

Figure 8: **AdapTok performs adaptive content-aware tokenization.** From the top to the bottom, AdapTok allocates more tokens as scene complexity and motion increase. Blue bars represent the token counts used per block.

## E  THE USE OF LARGE LANGUAGE MODELS (LLMS)

We used LLMs solely to assist with grammar correction of the submission. All research ideas, experiments, and analyses presented in this work were conceived, conducted, and verified by the authors.

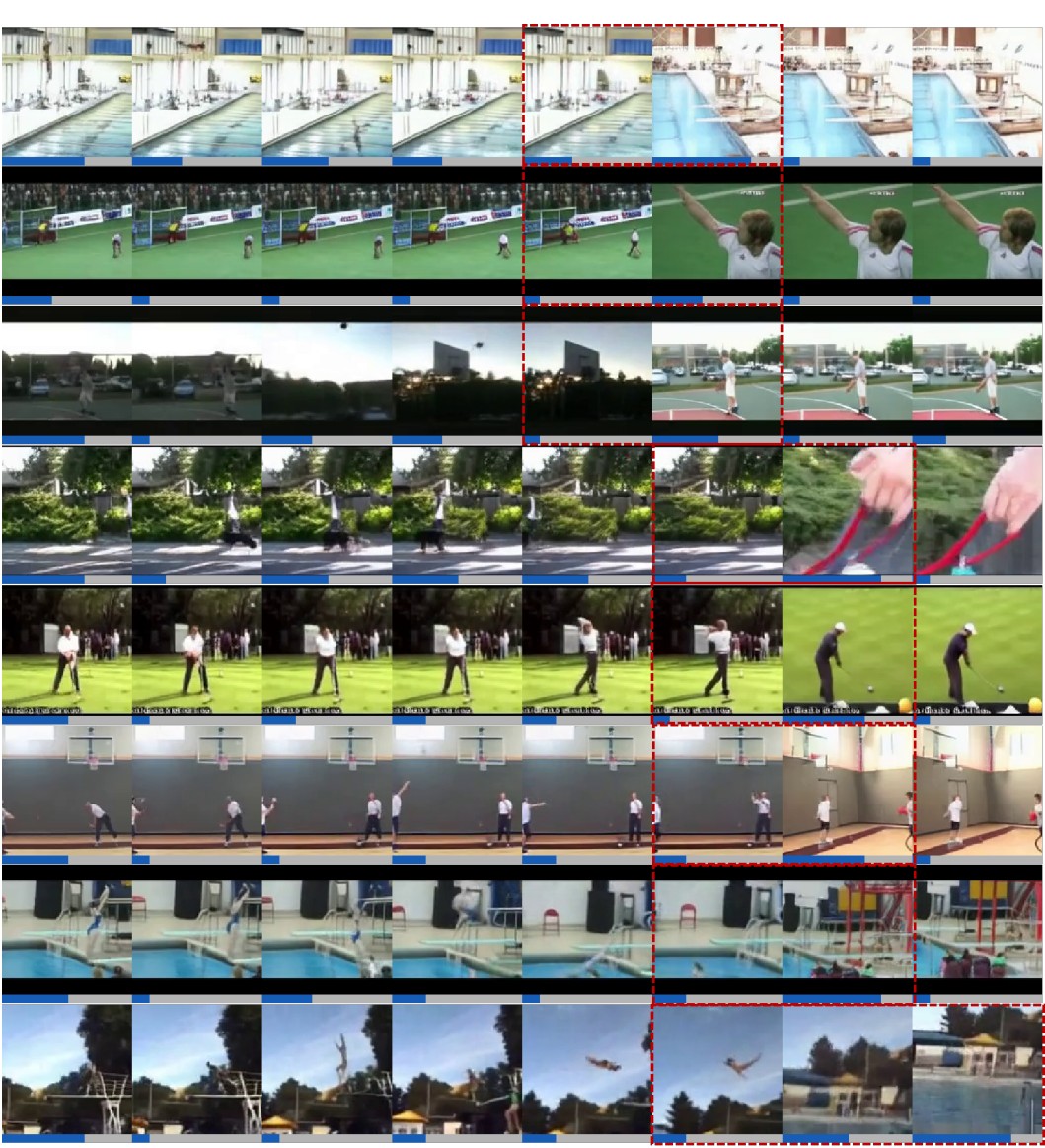

Figure 9: **AdapTok adaptively allocates tokens based on temporal dynamics.** Red boxes highlight the scene transitions, where AdapTok allocates more tokens to capture important temporal changes.

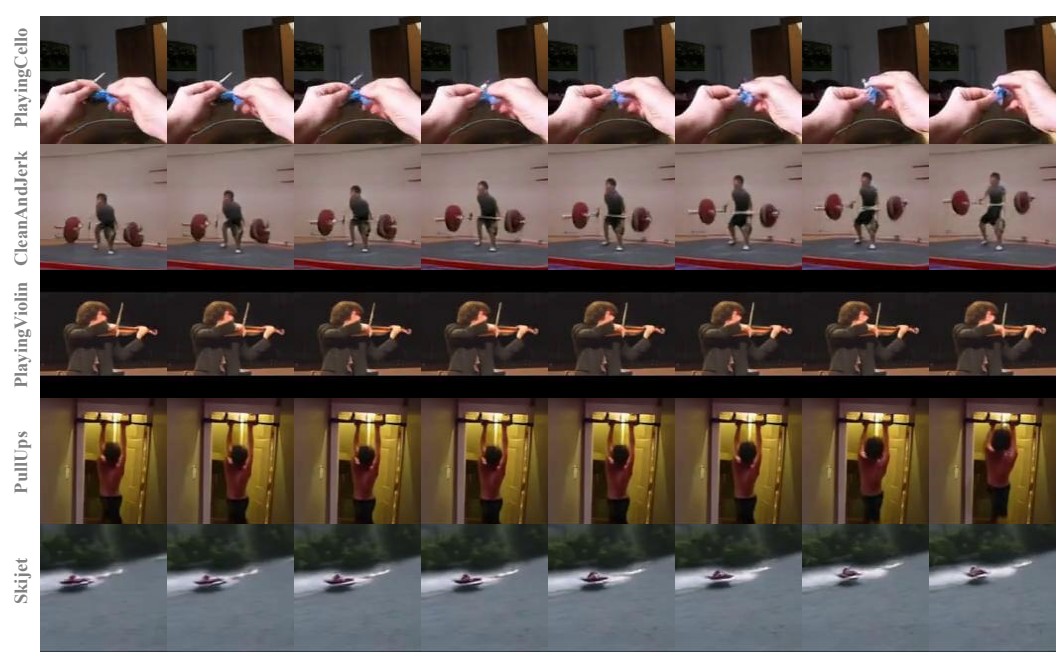

Figure 10: **Class conditional video generation on UCF-101.**

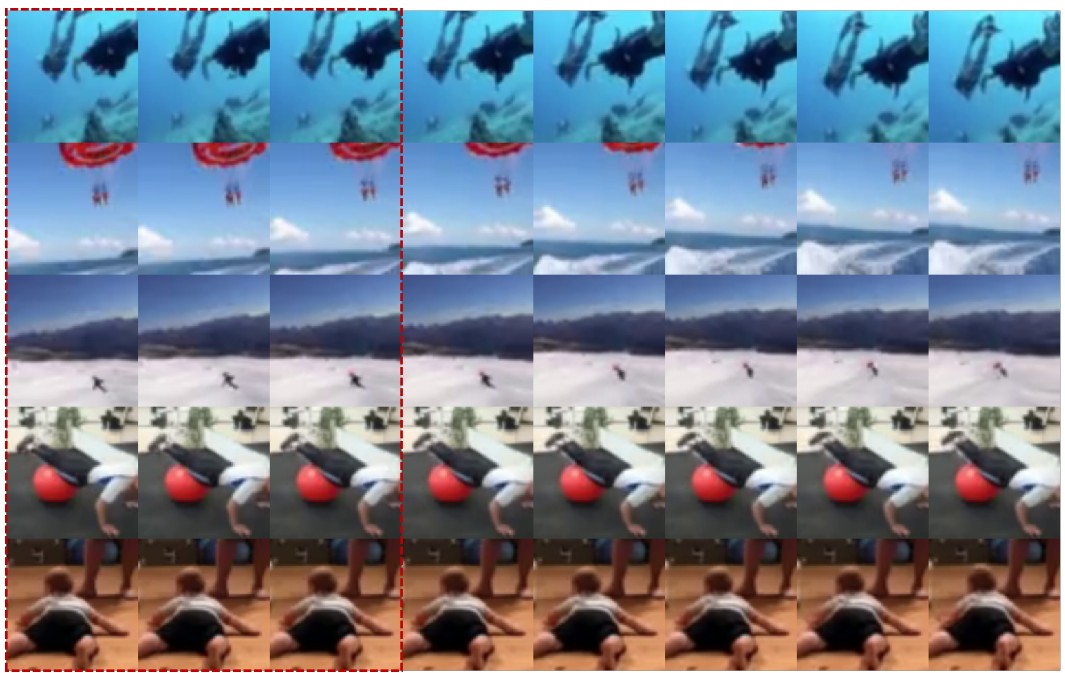

Figure 11: **Frame prediction results on Kinetics-600.** Red boxes represent the condition frames.

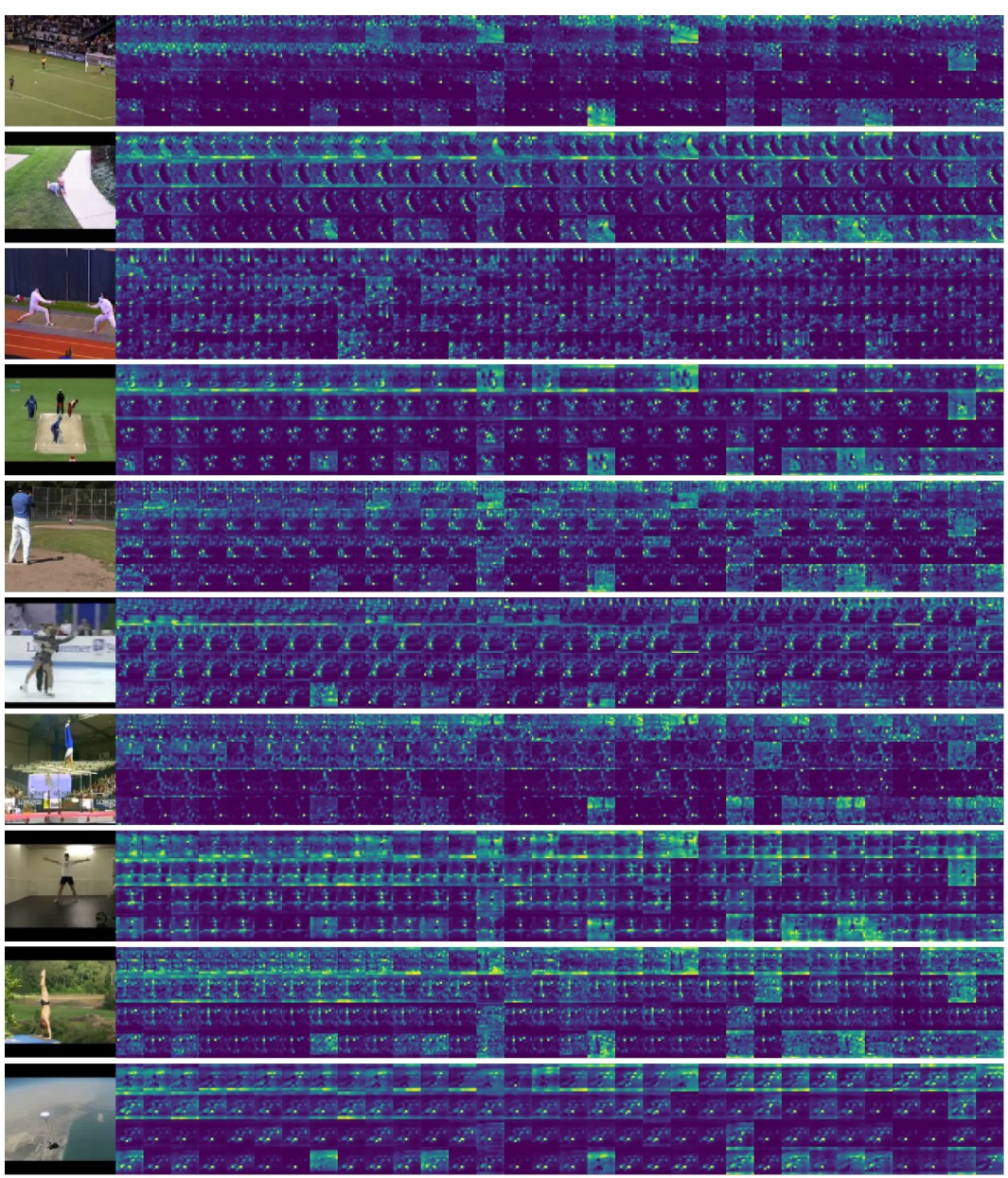

Figure 12: **Additional attention maps for latent tokens.** Each map shows how a token attends to spatial patches, ordered from top-left to bottom-right. Early tokens attend broadly to capture global context, while later ones focus on local details.

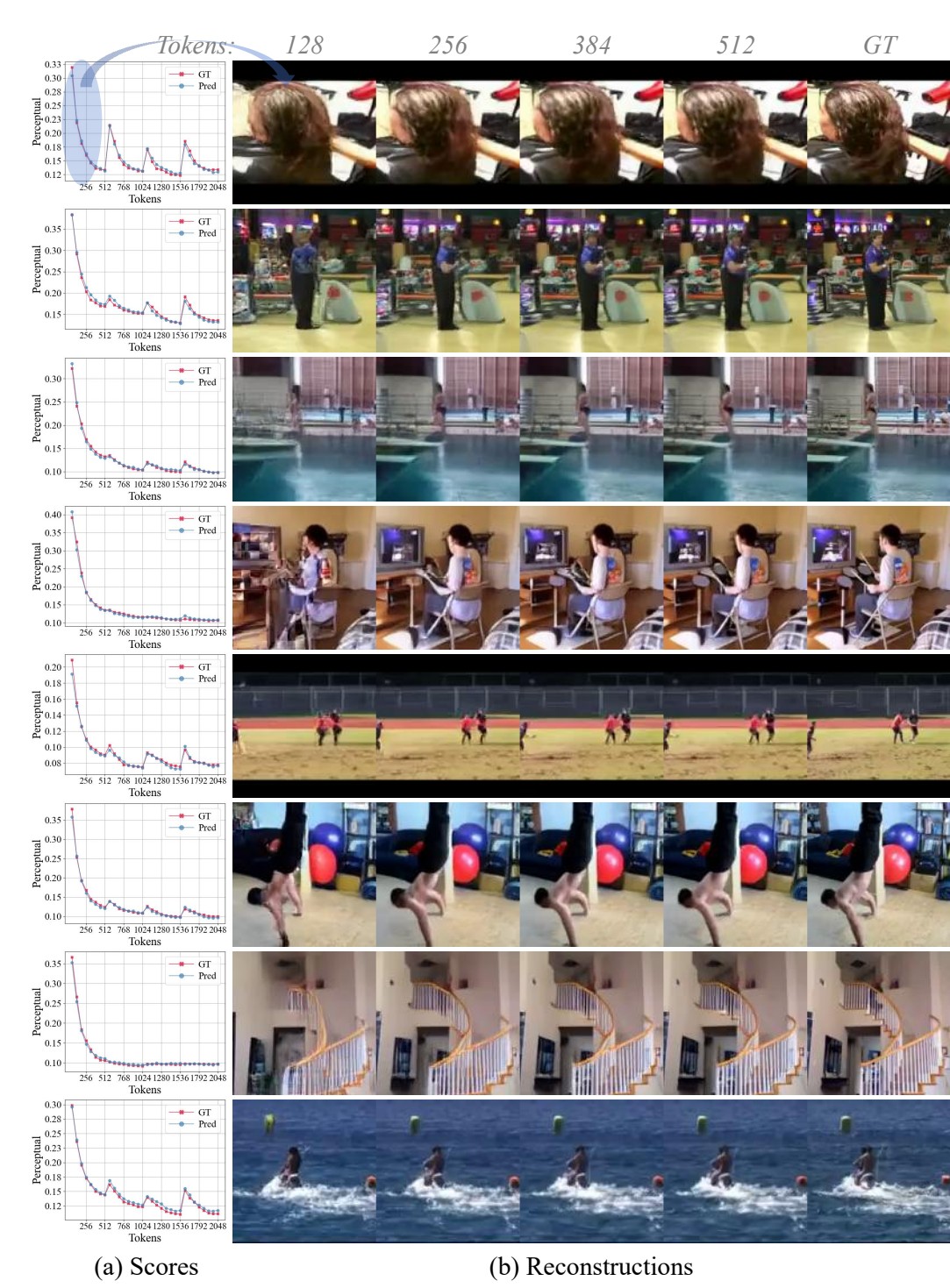

(a) Scores  (b) Reconstructions

Figure 13: **The visualization of scorer predictions and corresponding reconstruction results.**
(a) Scorer predictions. Red curves represent the GT scores, while blue denotes the predictions. A total of 2048 tokens are divided into 4 blocks. The x-axis represents the index of the last selected token for each block. (b) Corresponding reconstruction results of frames from the first block under different token counts.

