# OpenReview forum: "Learning Adaptive and Temporally Causal Video Tokenization in a 1D Latent Space"
_ICLR.cc/2026/Conference — ICLR 2026 Conference Withdrawn Submission_

### Official Review · Reviewer_f9qK · 2025-10-26

**Soundness:** 3
**Presentation:** 3
**Contribution:** 3
**Rating:** 2
**Confidence:** 4

**Summary:**

This paper presents AdapTok, an adaptive temporal causal video tokenizer designed to flexibly allocate tokens based on video content. The framework integrates a block-wise masking strategy for training, a block causal scorer for predicting reconstruction quality under varying token budgets, and an ILP-based token allocation strategy for inference. AdapTok claims three key features: temporal causality, a 1D latent token space, and adaptive token allocation. Experimental results on UCF-101 and Kinetics-600 demonstrate improved video reconstruction and generation performance compared to baselines like ElasticTok and Cosmos-Tokenizer.

**Strengths:**

1. The integration of a block causal scorer and ILP-based IPAL strategy addresses the limitation of fixed token budgets in prior work (e.g., ElasticTok), enabling global optimization of token usage across samples and temporal dynamics. This is supported by strong empirical results showing Pareto optimality between performance and token count.

2. By enforcing causal attention across blocks and decoupling token allocation from spatial structure, AdapTok supports online streaming processing and avoids spatial bias.

3. The authors validate AdapTok on both reconstruction and generation tasks, with ablations and comparisons to state-of-the-art causal tokenizers. The 11× lower inference latency compared to ElasticTok further highlights practical utility.

**Weaknesses:**

The manuscript exhibits several critical limitations in its methodology, theoretical justification, and experimental rigor, which undermine the validity of its claims and the interpretability of its contributions. Below is a detailed expansion of these weaknesses:

1. Oversight of MLLM Video Tokenization Literature.
The authors position AdapTok as addressing a "gap" in efficient video tokenization, yet they overlook a rich body of work on video tokenization in multimodal large language models (MLLMs), where token efficiency and causality are already core focus. Review [1] provides a detailed overview of compression methods for video tokens in MLLM. The paper lacks an analysis of the connections and differences between video token compression methods in video generation and video understanding. Additionally, in the field of video generation, there have been some studies [2,3] focused on token compression. The paper also lacks discussion of these studies. The oversight weakens the paper’s contribution narrative, as readers cannot assess whether AdapTok advances beyond existing solutions or merely repurposes established techniques.

2. Unsubstantiated Design of "1D Latent Token Space".
The second key feature, "1D latent token space," is presented as a core innovation but lacks theoretical justification and empirical validation. The authors claim 1D tokens enable "evenly distributed information density," but they do not define "information density" or explain how 1D structuring achieves this. There is no comparison between 1D and 2D token spaces. Without ablating the token dimension (e.g., testing a 2D variant of AdapTok), the authors cannot prove that 1D is superior.

3. Weak Enforcement of Temporal Causality.
Despite emphasizing causal modeling, AdapTok only applies causal attention between blocks, using full attention within blocks. With small K (number of blocks, K=4) and large M (tokens per block, M=512), the causal constraint is minimally enforced. Full attention within blocks allows tokens to attend to future frames within the same block. The choice of K=4 and M=512 is not justified. Why not using larger K? The authors provide no ablation on K and M.

4. Critical Gaps in Adaptive Scorer Analysis.
The Adaptive Scorer, framed as the "core contribution," suffers from insufficient validation, making its role and necessity unclear. (1) The scorer is claimed to "predict reconstruction quality under varying token budgets," but there is no data on its prediction error. For example, if the scorer estimation is not accurate, the subsequent ILP-based allocation would rely on noisy inputs, rendering the adaptive strategy ineffective. The authors need to report metrics like MAE between predicted and actual FVD/PSNR to justify the scorer’s reliability.
(2) Optimal token lengths per block is not analyzed. If the assigned token numbers are nearly identical across blocks, the adaptive allocation reduces to fixed budgeting, negating the need for a scorer. The authors must visualize token number distributions across diverse videos to demonstrate variability and justify the scorer’s utility.

[1] Shao, Kele, et al. "When tokens talk too much: A survey of multimodal long-context token compression across images, videos, and audios." arXiv preprint arXiv:2507.20198 (2025).

[2] Li, Xirui, et al. "Vidtome: Video token merging for zero-shot video editing." Proceedings of the IEEE/CVF Conference on Computer Vision and Pattern Recognition. 2024.

[3] Wu, Haoyu, et al. "Importance-based token merging for efficient image and video generation." Proceedings of the IEEE/CVF International Conference on Computer Vision. 2025.

**Questions:**

1. Can you provide concrete evidence that decoupling from spatial structure improves information density? How does the 1D design specifically mitigate the "spatial region imbalance" issue in ElasticTok?

2. Since block-wise causal attention is only applied between blocks, have you tested larger K (e.g., K=16) and smaller M to strengthen causal constraints?

3. What is the prediction error (e.g., MAE) of the scorer’s perceptual loss estimates? Could you show the distribution of token counts across samples/blocks?

---

### Official Review · Reviewer_QuMR · 2025-10-26

**Soundness:** 3
**Presentation:** 3
**Contribution:** 3
**Rating:** 4
**Confidence:** 5

**Summary:**

This paper proposes AdapTok, an adaptive, temporally causal video tokenizer operating in a 1D latent space. Its core objective is to enable sample-wise and temporally dynamic token allocation under a global token budget.

Experiments show that AdapTok significantly outperforms existing causal video tokenizers on UCF-101 and Kinetics-600, achieving a remarkably low video reconstruction FVD of 28 with only 512–2048 tokens. It also achieves state-of-the-art performance in downstream video generation tasks, including class-conditional generation and frame prediction.

**Strengths:**

1. Clear and practically relevant problem formulation: Video data exhibits substantial spatiotemporal redundancy, and fixed-length tokenization is inefficient. AdapTok is the first to achieve globally optimal adaptive token allocation within a causal, 1D latent space, aligning well with real-world demands for efficient video modeling.

2. Novel and cohesive technical design: The integration of block-wise tail dropping during training, a block-causal scorer, and ILP-based inference (IPAL) forms a complete and elegant pipeline that balances training flexibility with inference optimality.

3. Significant efficiency gains: Compared to ElasticTok, AdapTok reduces inference latency by 11×, and the IPAL component incurs only ~15% overhead—demonstrating strong potential for real-world deployment.

**Weaknesses:**

1. Lack of fair comparison with non-causal adaptive methods: Without comparing against high-performing non-causal tokenizers (e.g., MAGVIT-v2) under the same token budget, it remains unclear whether the causal constraint incurs a performance penalty.

2. Limited novelty in sampling strategy: The block-wise tail-dropping mechanism appears conceptually similar to prior works such as DC-AE 1.5 and FlexTok, which somewhat weakens the claimed technical novelty.

3. Scalability concerns regarding ILP: The paper does not evaluate or theoretically justify the efficiency of the ILP solver under more demanding scenarios (e.g., high-resolution or long videos with 240+ frames). Practical scalability remains unverified.

**Questions:**

1. In Table 4, the PSNR drops drastically to 14.38 when the block-mask sampler is disabled (with 512 tokens). This is highly counterintuitive—typically, using all tokens (i.e., no sampling) should yield the best reconstruction. Can the authors explain this anomaly?

2. How does AdapTok compare against CNN-based video VAEs (e.g., Wan VAE, WFVAE) in terms of reconstruction quality and efficiency?

3. Table 5 reports dramatically faster inference than ElasticTok. Given that IPAL involves solving an integer program, which is typically costly, what architectural or implementation choices enable such high speed?

4. In theory, non-causal tokenizers should achieve better reconstruction since they leverage full spatiotemporal context. Yet AdapTok (causal) outperforms existing methods. Is this due to the 1D latent design, the adaptive allocation, or other factors?

5. The scorer relies solely on perceptual loss (LPIPS). Could a multi-objective scorer (e.g., balancing LPIPS, PSNR, motion fidelity) improve robustness? Why was a single metric chosen?

---

### Official Review · Reviewer_ekbf · 2025-10-31

**Soundness:** 2
**Presentation:** 3
**Contribution:** 2
**Rating:** 4
**Confidence:** 4

**Summary:**

This paper proposes an adaptive framework (AdapToK) designed for video tokenization. AdapTok features temporal causality, a 1D latent token space, and a flexible adaptive token allocation strategy simultaneously. The authors further propose a causal scorer and a corresponding adaptive token allocation strategy to dynamically adjust token allocation across different samples.

**Strengths:**

1. The paper is well organized.

2. The topic of encoding different frames within the same video using varying numbers of tokens is worth exploring in the research community. This paper targets this important problem.

3. The paper provides a lot of figures to help reviewers better understand of the proposed method.

**Weaknesses:**

There are some concerns and questions about this paper:

1.	The authors mention in the introduction that VAEs need to possess the characteristic of temporal causality, but a significant drawback of this characteristic is error accumulation. How do the authors address this problem?

2.	I also have some questions about the 1-D latent token space characteristic. It's well known that a 1D latent space is very suitable for sequences like speech, but for non-1D input signals such as images and videos, a 1D latent space is not an optimal choice. Specifically, current mainstream image VAEs and video VAEs do not use a 1D latent space. Although we will expand the sequence into a 1D sequence in subsequent Transformer calculations, we will introduce RoPE positional encoding.

3.	In visual AR models, there's a current trend towards using continuous feature spaces rather than discrete feature spaces, as seen in the following works, MAR [1] and NOVA [2]. I'm curious about the authors' reasons for choosing discrete feature spaces.

4.	In the introduction, the authors mention that to represent a video using different numbers of tokens, they used a method of randomly discarding tokens. This seems like a very straightforward approach. Some token merging or pruning methods discard unimportant tokens. What advantages does the authors' method have compared to these methods?

5.	When evaluating reconstruction performance, the authors omitted another important metric, SSIM. Please add this metric. FVD measures the distance between two distributions, and I don't think it's very accurate for evaluating reconstruction performance.

6.	Could the author provide some comparisons between the original video and the reconstructed video?

**References:**

[1] Autoregressive image generation without vector quantization, NeurIPS 2024

[2] Autoregressive video generation without vector quantization, ICLR 2025

**Questions:**

Please see above. Please provide the SSIM results for reconstruction performance. Do you think that using your proposed VAE in T2V or I2V applications can outperform some current leading models, such as Wan2.1?

---

### Official Review · Reviewer_ZnuW · 2025-11-01

**Soundness:** 3
**Presentation:** 3
**Contribution:** 3
**Rating:** 6
**Confidence:** 4

**Summary:**

This paper proposes a novel video tokenizer that encodes videos into variable numbers of tokens adaptive to their content complexity. To achieve this, it designs a block-wise masking strategy to randomly drops tokens during training, and a scorer to predict reconstruction quality given certain number of tokens. During inference, it proposes an Integer Linear Programming procedure to estimate how long the sequence needs to be generated. The proposed models are evaluated on UCF-101 and Kinetics-600 datasets following prior work's setup, and reach leading performance with a more flexible scheme of token allocation.

**Strengths:**

- This work is well motivated with the purpose of modeling videos using variable numbers of tokens given its content complexity.

- The proposed tokenizer is well designed to dynamically drop out tokens in block level. The generator is also able to generate variable length of latent with <EOB> tags.

**Weaknesses:**

- Table 1 doesn't include the model size (params) of the tokenizers. Table 2 doesn't indicate which token number configuration is used from Table 1 (I'd assume it's the 2048 tokens one). Some most recent methods are not included, e.g. [1]. It's OK if the final performance doesn't surpass the previous SOTA, as long as it reaches comparable performance with more efficient models/tokens etc., as highlighted by the motivation of this paper (e.g. average generation tokens across all categories, minimum tokens to generate a simple video, etc.)

- Although no special design is needed for the generator, it's still better to highlight the dynamic generation process of variable token length in the figure (with the Integer Linear Programming).

- Supplementary files are missing and the temporal quality is hard to assess.

[1] LARP: Tokenizing Videos with a Learned Autoregressive Generative Prior. ICLR 2025.

**Questions:**

- For class-conditioned datasets like UCF and Kinetics, in many cases the class category has largely indicated the video content complexity (e.g. indoor/close-up videos are simpler than outdoor ones with more objects). How might class-adaptive token allocation perform? i.e. The encoder (and maybe decoder) takes in the class embedding, and the token allocation is also determined only by the class. In this way the same scorer might be able to be reused when training the generator as the class is also provided, while currently a separate Integer Linear Programming is needed to determine how long of the sequence to generate.

---

### Note · Authors · 2025-11-13

I have read and agree with the venue's withdrawal policy on behalf of myself and my co-authors.